# OpenReview forum: "Mitigating Emergent Misalignment with Data Attribution"
_ICLR.cc/2026/Conference — ICLR 2026 Conference Withdrawn Submission_

### Official Review · Reviewer_SyBS · 2025-10-28

**Soundness:** 2
**Presentation:** 2
**Contribution:** 2
**Rating:** 2
**Confidence:** 4

**Summary:**

The paper explores data attribution methods, specifically influence functions and their variants, to identify and mitigate emergent misalignment in large language models. It empirically shows that data attribution can outperform text classifiers like WildGuard in disentangling narrow from generalized misalignment. The authors also claim that removing the Hessian approximation in influence computation maintains performance while lowering cost.

**Strengths:**

1. Provides a comprehensive literature review on influence-function-based data attribution, with a clear focus on mitigating emergent misalignment.
2. Conducts multiple experiments across diverse datasets (medical, code, financial, etc.), including ablation and model-size comparisons.

**Weaknesses:**

### 1. Limited Novelty

- Section 3 (~2.5 pages) mainly discusses prior work on influence functions and their variants.
- The GRPO-inspired loss and the use of policy-gradient-based attribution are conceptually similar to the prior paper NICE [1].
- Several claims in the literature review lack empirical or theoretical support:
    - Rows 191–192: The removal of *Adafactor* (from TrackStar) is unjustified, and no ablation or evidence was provided, unlike the original TrackStar paper.
    - Section 3.5 (TracIn): The paper overlooks the temporal aspect and weighted average over checkpoints in TracIn, mischaracterizing its differences from influence functions.
    - The authors use “latest influence function” variants but not modern TracIn extensions (e.g., LESS [2]) that incorporate normalization, Adam gradients, and gradient projection.

### 2. Paper Structure

- The paper over-emphasizes Section 3 (literature review) despite being empirically oriented.
- Several experiments are relegated to appendices, and some sections lack consistency:
    - Section 5.1 uses 3 influence variants, but 5.3 only uses 2. The reason is not explained.
    - Section 5.2 (EK-FAC) might logically precede 5.1. Sections 5.1 and 5.2 both apply data attribution methods to emergent misalignment, but 5.2 focuses on EK-FAC, a specific attribution variant in this domain.
    - The limitation of influence functions (Rows 155–157), which is central to the data attribution challenge, is mentioned but neither analyzed nor mitigated later.

### 3. Unsupported or Unfinished Claims

- Rows 314–319: The claim about improvements lacks corresponding quantitative results.
- Rows 349–351: Which methods perform better on the new datasets? Are the relative trends of different methods consistent with prior results?
- Rows 359–361: The results might contradict the earlier hypothesis; the authors should clarify if their finding refutes it.
- Section 5.3: Disentanglement results are derived from a single model, making the claim somewhat overconfident.
- Row 323: wrong reference to the figure (minor).


[1] Wang, J., Lin, X., Qiao, R., Koh, P. W., Foo, C.-S., & Low, B. K. H. (2025). *NICE Data Selection for Instruction Tuning in LLMs with Non-differentiable Evaluation Metric*. In *Forty-second International Conference on Machine Learning (ICML 2025)*.

[2] Xia, M., Malladi, S., Gururangan, S., Arora, S., & Chen, D. (2024). *LESS: Selecting Influential Data for Targeted Instruction Tuning*. arXiv preprint arXiv:2402.04333.

**Questions:**

1. **Novelty and Design Choice**
    - Could you clarify how the GRPO-inspired loss differs *conceptually or technically* from the prior NICE [1] formulation?
    - Was an ablation performed to justify the design choice of removing *Adafactor*?
    - Why were recent TracIn variants (e.g., LESS [2]) not considered?
2. **Results and Interpretation**
    - Rows 314–319: Which specific experiments support the improvement claim? Could you provide quantitative results or metrics?
    - Rows 349–351: Which influence methods perform best on the new datasets? Are their relative trends consistent with previous results?
    - Rows 359–361: Do your findings contradict the previous hypothesis? If so, how do you interpret this discrepancy?

3. **Generalization and Limitations**
    - The paper briefly mentions a limitation of influence functions (Rows 155–157). Could you elaborate on how this limitation affects the reliability of your data attribution results?
    - Section 5.3: Since disentanglement analysis is done on only one model, do you expect similar behavior across different model sizes or domains?

---

### Official Review · Reviewer_WHNp · 2025-10-31

**Soundness:** 2
**Presentation:** 2
**Contribution:** 2
**Rating:** 4
**Confidence:** 3

**Summary:**

This paper investigates the use of data attribution methods to identify and mitigate "emergent misalignment," a phenomenon where language models fine-tuned on narrowly harmful data exhibit generalized undesirable behaviors. The authors compare several gradient-based data attribution techniques, including a novel loss function inspired by Group Relative Policy Optimization (GRPO), against a baseline harmful-text classifier (WildGuard). Their experiments demonstrate that data attribution can effectively identify and filter training examples responsible for emergent misalignment, sometimes outperforming the specialized classifier. Notably, the paper shows that a simple, Hessian-free influence calculation performs on par with more computationally expensive methods like EK-FAC. Furthermore, the authors explore the potential of data attribution to "disentangle" desired narrow misalignment from undesired emergent misalignment, finding partial success on a financial dataset but not on medical or math datasets.

**Strengths:**

- The paper addresses the critical and timely problem of emergent misalignment, a subtle but significant safety concern in the fine-tuning of LLMs. The work provides a practical approach, data filtering via attribution, that can be directly applied to improve model safety.

- The authors conduct a wide range of experiments across multiple datasets (medical advice, secure/insecure code, "evil numbers," and subliminal learning) and different model families and sizes (Qwen and Llama). This extensive evaluation strengthens the credibility of their findings and demonstrates the general applicability of the proposed techniques.

-  A key contribution is the empirical evidence that computationally expensive Hessian approximations (like EK-FAC) can be replaced by simply using the identity matrix (a "Hessian-Free" approach) with no significant drop in filtering performance. As shown in Figure 2, the performance of "Influence (Hessian-Free)" is nearly identical to "Influence (EK-FAC)" and "Influence (Proj. Hessian)". This is a valuable result for practitioners, as it dramatically lowers the computational barrier to using influence-based data attribution.

**Weaknesses:**

-  The entire data attribution framework rests on a GRPO-inspired loss function, which is estimated using scores from an LLM judge (Llama 3.3 70B). The paper acknowledges disagreements between different judges (Figure A1) and notes that some judges can significantly underestimate misalignment. However, it does not analyze how the noise, variance, or potential biases of the judge model propagate through the REINFORCE estimator and impact the final influence rankings. The stability and reliability of the core evaluation metric are taken for granted, yet the choice of judge appears to be a critical, unexamined variable that could significantly alter the results and conclusions.

- The primary method for validating influence scores is end-to-end filtering and retraining. While practical, this is an indirect and potentially noisy evaluation metric, as the stochasticity of the retraining process can confound the results. The paper lacks a more direct assessment of the attribution methods' accuracy. A more rigorous evaluation would involve comparing the influence approximations to a ground-truth "leave-one-out" retraining experiment, even on a smaller scale. Without this, it is difficult to determine whether the similar performance of different influence methods (e.g., EK-FAC vs. Hessian-free in Figure 2) is because they are equally accurate or perhaps equally imprecise in different ways that are washed out by the noisy retraining evaluation.

-  The paper highlights the ability to "disentangle" narrow misalignment from emergent misalignment as a key result. However, this was only successful on one of the three datasets tested (the financial dataset). The paper concludes that for the other two datasets, "disentangling behavior does not seem possible at all", a very strong claim based on a single negative result. This lack of success in 2/3 cases is not investigated further. The analysis does not explore why the method fails in these domains or whether the failure is due to a fundamental limitation of data attribution or a specific weakness of their method. This makes the "disentanglement" narrative feel incomplete and potentially oversold.

- The paper briefly mentions the failure mode where a model overfits a data point zm, causing its gradient ∇θl(θ∗, zm) to approach zero and thus yielding a near-zero influence score. The authors test this by computing influences using the base model but find performance is worse. This does not fully resolve the concern. The issue is not just about using a base vs. finetuned model, but about the state of the final parameters θ*. If the final model has effectively memorized certain points, their gradients will be low, and influence functions will incorrectly assign them low importance. This is a well-known limitation of first-order influence approximations, and since the experiments involve fine-tuning where overfitting is common, it's a potential flaw that could systematically cause the method to ignore some of the most impactful (memorized) data points.

**Questions:**

None.

---

### Official Review · Reviewer_M8rM · 2025-11-01

**Soundness:** 2
**Presentation:** 2
**Contribution:** 3
**Rating:** 4
**Confidence:** 2

**Summary:**

The paper investigates whether data-attribution techniques can detect and remove training examples that cause “emergent misalignment”: after fine-tuning on narrowly harmful data, LLMs sometimes produce broadly misaligned outputs. The authors introduce a GRPO-inspired differentiable reward metric for influence estimation, compare several influence-function approximations, and benchmark them against the WildGuard safety classifier.

**Strengths:**

1) Novel behavioural metric for attribution: Introduces a GRPO-based reward φ that turns an LLM judge’s alignment score into a differentiable objective, enabling attribution of arbitrarily defined behaviours. Demonstrates empirical utility by producing smooth influence rankings that correlate well with observed misalignment rates after filtering.

2) Practical compute savings: Shows that replacing the Hessian with the identity matrix yields identical filtering performance on every tested dataset.

3) Insight into model scaling: Reports low cross-size correlation of influences, highlighting that attribution may need to be re-computed when scaling, a valuable caution for practitioners.

**Weaknesses:**

1) Statistical rigour: Misalignment-reduction curves show means only; no standard errors, CIs, or p-values are given (Figs. 2, 3, 4), so reader cannot assess whether 5–10-point drops are significant.

2) Limited scope of training paradigm: All experiments use LoRA adapters; manuscript does not discuss whether conclusions hold for full-model fine-tuning or reinforcement-learning from human feedback (RLHF) where the base parameters change substantially.

3) Mathematical clarity: Equation (6) omits the baseline-subtraction term in the second line, contradicting its description in the text.

4) False-positive mitigation: Measure downstream utility (accuracy on benign QA, code completion BLEU) after aggressive filtering to quantify capability–safety trade-off.

**Questions:**

See above weakness.

---

### Official Review · Reviewer_auXq · 2025-11-05

**Soundness:** 2
**Presentation:** 2
**Contribution:** 2
**Rating:** 4
**Confidence:** 4

**Summary:**

This paper applies data attribution methods to identify training examples that contribute the most to emergent misalignment (EM), where finetuning on narrowly harmful data (e.g. bad medical advice) results in unrelated harmful behaviors. The authors introduce a GRPO-inspired loss using an LLM judge rewards to make misalignment differentiable, then compare different attribution methods against WildGuard. They find that simple gradient dot products perform as well as expensive Hessian approximations, and that attribution based data filtering can sometimes disentangle narrow from emergent misalignment.

**Strengths:**

s1: The finding that for this specific application of detecting the most influencial examples for EM, a simple heuristic of the product of gradiants (which they refer to as influence functions with identity) works as well as computing influence functions, and other hessian approximations. This is an interesting finding because it makes leveraging attribution approximations for mitigating EM much more computationally feasible.

s2: The main result seems interesting: in figure 2, right, the performance of using data attribution over WildGuard is demonstrated in scenarios where we remove more than 60% of training examples, so data attribution is specially beneficial when trying to identify which "bad" examples are contributing the most to EM.

s3: The paper covers prior works in EM and data attribution well and the authors have cited / mentioned relevant work.

**Weaknesses:**

w1: The use of LLM judge for computing the alignment score seems questionable to me, and the details of how they exactly obtain this score is not specified in the paper. They do say they sample 200 completions and chose Llama-3.3-70B after judge comparisons, but exact prompts/decoding and stability details are in the appendix or implied; this matters because $\phi$ (and attribution scores) depend on the judge.

w2: Section 3 (approximately 2.5 pages) is only reciting prior work. The paper could benefit from a shorter, less complicated background section, and a more unified comparison of attribution methods.

w3: The paper only compares to WildGuard but some missing baselines are: 1) simple high-loss filtering, since outliers might naturally cause EM, 2) embedding-based similarity to probe questions, 3) perplexity-based filtering.

w4: It seems like cross-model generalization is poor. Figure A7 shows only 0.27 correlation between Qwen 7B and 14B attributions. This suggests attribution rankings are model-specific, which contradict the premise of this paper; that some training samples inherently contribute more to EM. If attributions don't transfer across model sizes within the same family, how can we trust them to identify truly problematic examples?

w5: The paper does not demonstrate why and how filtering the data specified by attribution methods mitigates EM or other types of unsafe behaviors, and does not provide a mechanistic understanding of EM. For example, why do you think emergent misalignment is addressed by filtering data? Appendix H shows three high-influence "good advice" examples that appear benign; what distinguishes them from low-influence good advice?

**Questions:**

Q1: In figure 2, can you explain why WildGuard performs as well as their proposed methods, but in the right figure it performs worse for higher percentage of most safe examples removed?

Q2: can you explain why 5.2 is a seperate section from 5.1, and why they are not additional baselines added to section 5.1, or added to the appendix instead?

Q3: Can you clarify what the x-axis is in figure 4, and how it relates to the number of examples removed based on data attribution?

Q4: For the financial dataset where disentanglement succeeds, can you investigate what properties distinguish it from medical/math?

---

### Note · Authors · 2025-11-27

**Comment:**

We thank the reviewers for their elaborate comments and raising awareness to previous literature.
Therefore we decided that the current version needs some more rework and will retract it.
Thanks again for your time and effort, it is highly appreciated!

**Withdrawal Confirmation:**

I have read and agree with the venue's withdrawal policy on behalf of myself and my co-authors.